# Roma Religion: 1775 and 2018 Compared over Time

**DOI:** 10.3390/ijerph191811645

**Published:** 2022-09-15

**Authors:** Michal Kozubik, Daniela Filakovska Bobakova, Martina Mojtova, Miroslava Tokovska, Jitse P. van Dijk

**Affiliations:** 1Department of Social Work and Social Sciences, Faculty of Social Sciences and Health Care, Constantine the Philosopher University in Nitra, 949 74 Nitra, Slovakia; 2Department Community & Occupational Medicine, University Medical Centre Groningen, University of Groningen, 9713 AV Groningen, The Netherlands; 3Olomouc University Social Health Institute, Theological Faculty, Palacky University Olomouc, 771 11 Olomouc, Czech Republic; 4Graduate School Kosice Institute for Society and Health, Faculty of Medicine, Safarik University, 040 01 Kosice, Slovakia; 5Department of Health Psychology and Research Methodology, Faculty of Medicine, Safarik University, 040 01 Kosice, Slovakia; 6Department of Health and Exercise, School of Health Sciences, Kristiania University College, Prinsens Gate 7–9, 0152 Oslo, Norway

**Keywords:** Roma, religion, comparison, 18th century, 21st century, Slovakia

## Abstract

The objective of the present study was to compare the religiosity of the Roma in the 18th century with the present. In 1775 and 1776, Samuel Augustini ab Hortis detailed the way of life of the Roma community in the Austro-Hungarian monarchy in his work “*Von dem Heutigen Zustände, Sonderbaren Sitten und Lebensart, Wie Auch von Denen Übrigen Eigenschaften und Umständen der Zigeuner in Ungarn*” (On the Contemporary Situation, Distinctive Manners and Way of Life, as Well as the Other Characteristics and Circumstances of Gypsies in Greater Hungary). A detailed content analysis of the part of his work dealing with religion was performed. Subsequently, in 2018, field research was conducted in the environment in which Samuel Augustini lived and worked. It involved six key informants, each representing a different municipality. Data collection was carried out over two periods: in the summer months of 2012–2013 and the winter period of 2018–2019. After the interviews with the key informants, more than 70 participants were included in semi-structured interviews through snowball sampling, and another 40 participants were included in two focus groups. The data was evaluated and content analysis was used to process the data. The findings confirm that both in the past and the present, the Roma community adopted the dominant religion of the host country. In the studied environments, the activities of the majority, present then and now in the Catholic Church, failed, and various other missionary movements, such as the Maranatha Mission, came to the fore. Membership in new religious movements resulted in social changes in marginalized Roma communities. However, they may not have only had positive effects. Various effects of their activities may be studied in the future.

## 1. Introduction

A general picture of religion among the Roma people is known. Roma have generally adopted the religion of the countries where they live and are strongly influenced by the religious beliefs of the majority [1]. The differences between Roma sub-ethnic groups may vary and have been very well described [2,3]. Roma religious denominations in Slovakia have also been described in detail. One study confirmed that the influence of religious missions leads to positive social changes and, under certain conditions, also to social inclusion [4]. However, there is a lack of insight into how the religiosity of the Roma has evolved and changed compared to the past. Herein, we focus on a description of Roma religion in the past, during the reign of Maria Theresa in Austria-Hungary, as captured by Samuel Augustini ab Hortis [5,6], in 1775–1776, and its comparison with the present form.

“The relationships between religion and the health of individuals and populations has become increasingly visible in the social, behavioral, and health sciences. Recent research has validated the multidimensional aspects of religious involvement and investigated how religious factors operate through various bio-behavioral and psychosocial constructs to affect health status through proposed mechanisms that link religion and health” [7]. Religious beliefs and behavior may have a negative impact (e.g., fanatical violence, mortifying asceticism, and oppressive traditionalism) as well as a positive influence on people’s physical and mental health [8] (the role of religious practices in personal health, the impact of social ministries on community health, and the complementarity of religious ideas of salvation with medical conceptions of health in contemporary conceptions of human wellbeing) [9]. For Roma communities, there could be both pros and cons of religiosity. The Roma are Europe’s largest ethnic minority [10] and are not a homogeneous group of people [11]. After the collapse of communism in Romania, the post-Soviet republics, and the Slovak Republic, different religious denominations came from abroad to these countries to find new followers [1]. Roma do not follow a single faith; rather, they often adopt the predominant religion of the country where they are living. Some Roma groups are Catholic; some are Muslim, Pentecostal, Protestant, Anglican, or Baptist [12]. Just as the religion may be perceived differently by the population in individual countries, the religiosity of different sub-ethnic Roma groups may be diverse, too. Religious and social mobilization may grow from local missions to transnational alliances. For instance, Roma mobilization in Finland continues to have a religious undertone, especially in the contemporary transnational humanitarian work conducted by Finnish Roma missionaries among Roma communities in Eastern Europe [13].

After 1990, groups of individuals supported from abroad started operating in almost every district town in Slovakia and gradually established congregations of the Pentecostal movement. In the new places of activity, the denominations began with intensive missionary work that affected the Roma, too. Their missionary work was quite effective, and those small, relatively dynamic communities of converts were able to respond flexibly to changes and to the needs of working with the Roma in settlements, not only in the area of education but also in practical assistance in the field [14]. The ethnicity of the Roma minority in the Slovak Republic is interconnected with several categories. Soltesova [15] described it as a multi-layered identity. Her research in Slovakia has also revealed a multi-layered religiosity—a combination of older pre-Christian and new Christian concepts and ideas. Podolinska [16] mentions the concept of the *New Roma* as a de-ethnicized and historically constructed label with positive and non-ascriptive connotations. The *New Roma* concept offered to Roma by pastors is also likely to increase the potential of Roma to enter secondary and other kinds of networks within the mainstream society and allow them positive visibility at the mezzo-level of society. Around 2004 and 2005, Emil Adam, a Roma missionary from the Czech Republic, started working in (the Slovak) Spis region; later, he founded the Roma Maranatha Pentecostal Mission there. Over a few years, he managed to establish a number of strong local groups in the villages near the town of Spisska Nova Ves (Rudnany, Porac, Bystrany, Zehra, Markusovce) and in the vicinity of Poprad (Hranovnica, Vikartovce, Spissky Stvrtok). Now, several Roma pastors are trained in this church, who have extended the Mission to areas in the vicinity of Giraltovce and Humenne [4].

Although several research studies on the Roma religion have been conducted, there has been no comparisons of religious perceptions in a historical context. Therefore, we attempt to bring more knowledge to the area of religious expressions and customs in the Austro-Hungarian Empire, as described by Samuel Augustini (1729–1792), and to compare it with the current situation.

## 2. Methods

Samuel Augustini published his work during the reign of Maria Theresa in Austria-Hungary in 1775–1776. His summary work was called *Von den heutigen Zustande, sonderbaren Sitten und Lebensart, wie auch von den übrigen Eigenschaften und Umständen der Zigeuner in Ungarn*, i.e., “On the Contemporary Situation, Distinctive Manners and Way of Life, as Well as the Other Characteristics and Circumstances of Gypsies in Greater Hungary”. The introduction of his work was published in 1775 in the 20th issue of the magazine called *Kaiserlich Königliche allergnädigste Anzeigen aus sämtlichen Kaiserl. königl. Erbländer*. During the course of the year, which was the fifth year of the magazine, it published 25 of Augustini’s parts. In 1776, the journal published 14 more parts, for a total of 39 parts. As it was not written in the form of a book, the text as a whole remained unnoticed for many years [6]. In the first part of our study, we mapped the area of religiosity by content analysis of Augustini’s work. We conducted our study in the same part of eastern Slovakia where Augustini lived and worked more than 250 years ago. Furthermore, we used the same qualitative research design to get as close as possible to Augustini’s perspective in the study. Work has already been published on housing and eating, illnesses and death, and the social structure of the Roma settlement [17,18,19]. Document analysis is a systematic research technique that uses both printed and electronic materials as its source, analyzing them in depth to extract information and indications related to the study objective [20]. It is important to note that document analysis is a process involving skimming (superficial examination), reading (thorough examination), and interpretation of content to provide answers to research questions [21].

### 2.1. Samples

In the first phase, we focused on Augustini’s work [5]. We used his verbatim transcript, which was published in German. A translation of the texts in Slovak was published by Urbancova in the early 1990s [6]. Augustini’s monograph draws on multiple sources, including mainly contemporary but also older literature. In addition, manuscripts and historical source material in which Augustini analyzed even the slightest mention of the Roma were included. Studying literature, he mainly used the large collection of the Evangelical Lyceum and Library in the nearby town of Kezmarok.

We carried out our field research in the north-eastern part of the Slovak Republic, in the Poprad district (Figure 1). Compared with the total number of Roma throughout Slovakia (8%), a greater proportion of Roma (30%) live in this region [19,22].

The reason for choosing the locality was the fact that Augustini himself lived in Spisska Sobota, which is now a part of the town of Poprad. In addition to this town, the villages of Hranovnica, Spisske Bystre, Kravany, Vikartovce, and Spisska Teplica were included in the study (Figure 2). We included all types of respondents: those integrated into the majority population (the town of Poprad and its suburb Spisska Teplica), those separated on the village periphery (Kravany, Vikartovce), and those segregated (Spisske Bystre, Hranovnica). The number of Roma inhabitants in these concentrations varies from 270 in Kravany to 1734 in Hranovnica [19,22].

More than 70 participants from Poprad and the neighboring villages participated in the study. In addition, two focus groups were conducted, each involving 20 participants. All participants confirmed their ethnicity and voluntarily consented to participate in the study in audio recordings.

### 2.2. Data Collection and Reporting

The starting point of our comparison was the content analysis of Augustini’s work. Augustini captured the area of religion in the chapter: *Von der Religion der Zigeuner,* “On the Religion of the Gypsies”. In his work, Augustini exclusively used the word *Zigeuner* (Gypsy). It was the only used term in the second half of the eighteenth century; therefore, in the descriptions related to the past, we preserve the ethnonym *Zigeuner*. Nowadays, however, the word Gypsy has a pejorative connotation both in Slovak and in English.

After the detailed analysis of the texts, we continued in the field research on the detection of the current form of religiosity in the environment of marginalized settlements. We decided for one key informant per village, who should come directly from the communities visited. “Key informants are those whose social positions in a research setting gives them specialist knowledge about other people, processes or happenings that is more extensive, detailed or privileged than ordinary people, and who are therefore particularly valuable sources of information to a researcher, not least in the early stages of a project” [23]. Then, staying at one of their places, we visited the nearby areas—the natural environment of families and informants. We obtained new information through snowball sampling and got to know new families [24]. Initially, data collection took place intensively in the summer months of 2012–2013. To ensure a higher validity of the information collected from the interviews, we conducted a shorter follow-up study in the winter months at the turn of 2018–2019.

Coding itself is a process of data analysis. We started analyzing the data to identify the phenomena and develop the characteristics and dimensions of categories. We rewrote all the records obtained from the semi-structured interviews. Afterwards, the transcriptions were analyzed through an open coding process. From the codes, four main categories were created: (1) “Change of life—acceptance of Jesus”, (2) “Miracles and testimonies”, (3) “Preaching and the Gospel”, (4) “Us and the others”. We tried to present them through personal statements of the participants. From our point of view, this helps to present a true image of the daily routine of the settlement dwellers. In our study, categorization is the process of grouping concepts that are linked to the same phenomenon. An important source of naming is the words and statements of the informants themselves. Our ethnographic field research had the goal of acting directly in the settlement environment and capturing sociocultural reality through the eyes of the inhabitants of marginalized communities.

### 2.3. Research Ethics

In our research, neither animals nor plants were studied. Human beings from 1775 to 1776 were not studied by us, and human beings in 2012–2013 and 2017–2019 were studied in line with the Helsinki Declaration. Before including participants, information about the study was given, and informed consent was obtained and archived through audio recordings. All participants were guaranteed that their information would be anonymized; they were also guaranteed the right to withdraw their data at any time before publication. 

## 3. Results

We intended to compare the areas of the Roma religiosity in two historical periods. The presentation of the results will therefore be divided into two parts. The first part will be a detailed analysis of Augustini’s work, which was presented in 1776. The second part will describe the current situation. We will present individual current phenomena through verbatim statements of the participants to highlight their authenticity. All citations in the paper have been translated from Slovak by the study authors.

### 3.1. Roma Religion before and in Augustini’s Period

Augustini not only describes the religion of the Roma in his time but also uses other literary sources and describes it in the past. He depicts the differences in the different parts of Austria-Hungary: in Transylvania (now: Romania) and in the northern part of the Austro-Hungarian Empire in the place where he lived and worked (Figure 3), namely Spisska Sobota, which is now a part of the district town of Poprad [5,6].

In the introduction to the chapter on Roma, he describes the religious denomination of the Roma living in Transylvania, then part of the Kingdom of Hungary, and the upper part of Hungary, where he lives. He stated that the Roma living in Transylvania professed the Greek religion. The Roma who joined the Catholic Church in Sibiu, a city in Transylvania, are an exception in this region. This was a large group of believers who had their own chaplain. The Roma inhabiting the territory where Hungarian was the lingua franca were divided into Catholics and Reformed, just as it was in the place where a Roma works. What these groups have in common, however, is that religion is just an externally demonstrated faith. As for their inner beliefs and faith, their knowledge of theology is absent. He literally states, “*With their hearts, they do not belong to one religion…*”.

Describing baptism, he also writes that the Roma baptize their children, but only because they live under Christian denomination and rule. Some inhabitants even accuse them of going from village to village with a newborn baby, repeating this scam for as long as they can. As an example, he cites the 1661 police regulation: “*Their (i.e., gypsy) children must not be baptized in any other place other than where they were born and when it is quite certain*”. Augustini states that Roma parents do not lead their children to prayer nor any other religious acts.

Augustini goes on to interpret an episode related to faith in the Resurrection. Roma parents lost their son, and when asked by a pastor if they believed their son would come back to life after death on the Resurrection day they replied, “*But this is a strange case of believing that a corpse, a lifeless body will come back to life and come from the dead! In our opinion, he will not come back to life, just like the horse from which we removed the skin*”. The author describes the Roma as people who live without worries about their future fate in eternity, because they do not think about it or do not believe in it. They rarely go to church.

In his conclusions, Augustini muses about the common presence of the Roma in a Church Service. He thinks it can merely be wished for, because even in such a holy place, they would not think about lifting their spirits. In his view, a joint worship service with the Roma could not continue with contemplation, because he could not be sure that something would not go missing; he thinks Roma might try to steal church treasures and sacred objects.

Such statements containing stereotypes and prejudices should be seen in the context of the time in which they were made. In other parts of Europe at the time of Augustini’s life, Roma were expelled, exterminated, and outlawed [6]. The reforms that affected the Roma in Austria-Hungary need to be assessed in the context of Maria Theresa’s and Joseph II’s reforms aimed at educating all the inhabitants of the state to make them equal citizens.

### 3.2. Roma Religion at Present

Two religious denominations among Roma have been identified in the north-eastern region of Slovakia, in the Poprad district where Augustini lived and worked about 250 years ago: the Roman Catholic and the Maranatha Mission. During our field research, besides the Roman Catholic Roma, we met with a newly appointed pastor and many active members of the Maranatha Mission. They received us warmly and were very happy and enthusiastic to talk about their relationship to faith, the Savior, and Jesus in their lives. Their statements may be categorized in the following areas [26]:“Change of life—acceptance of Jesus”Miracles and testimonies“Preaching and gospel”“Us and the others”.

#### 3.2.1. Change of Life—Acceptance of Jesus

Metaphorically, the Roma equate this transformation with the awakening given by Jesus Christ. Maranatha Mission members give up their vices and pleasures after “awakening”. The change resides in the discovery of a “living God” (due to the possibility of sensitive statements, we have omitted all data on the origin of informants) [26]:


*“When you pray, you have a relationship with God. You get wisdom. The Roma believe that Jesus exists. That is automatically when you look at the sky: what is it? Where has it come from? That must be someone higher above us. They know it is God, but they do not know Him. They do not know who God is or what He wants. I had idols and I have been saved for four years. I also had a painting, the Virgin Mary, Jesus Christ. Then I asked myself a question: If we have God in the house, how can we sin? God is looking at us, He is alive. Why have a picture of a God who does not save? I want a living God who talks, who gives me direction. I had idols, I lied, I drank, and I did not give any religion to the kids either. What good is God on our wall if we do not live like that? We must pray, we have a relationship with God…”. (1; m; age 35 y)*


Many Roma who became active members of the Maranatha Mission stopped smoking and drinking alcohol, and also stopped surrounding themselves with idols (holy pictures, statuettes of angels, etc.):


*“There used to be xxx (anonymous musical bodies, authors’ comment), and they create disorder here. People drink here, do other things, which is not good. It is not good for me. For others it may, maybe it is, but not for me”. (1; m; age 35 y)*


Joining the Mission is not easy for everyone. Some family members even perceive joining a Christian faith as a betrayal:


*“Our own family wanted to separate us for allegedly betraying the faith because we had taken down the pictures. So, when I drank, I was good, everyone loved me, but when I went into faith… not then, because I stopped drinking, I stopped sinning”. (2; f; age 46 y)*


Most often, the change is attributed to the miracles witnessed by the new believer. According to them, these testimonies are proof that God and Jesus Christ are still alive [26].

#### 3.2.2. Miracles and Testimonies

When we met with active Mission members, each of them was eager to tell us their own story—a miracle or a testimony. In each of the statements, there is a noticeable presence of *a white, a gadjo.* With their membership in the community and a radical transformation, the Roma feel like they are on an equal footing with the majority. That feeling was previously unknown to them. It may even be deduced from their statements that they feel smarter and more clever in many ways. Testimonies of experienced miracles consisted, for example, in the birth of a child whose mother was not recommended by doctors to have other children [26]:


*“… the doctors said that if we had a fourth child, the woman would die. They advised us to have an abortion. God does not want abortions. He who kills has sinned. The doctors said we could not have the fourth one. Our faith gave us such strength that we were not afraid, and we believed. When she went in the ninth month, I was praying in the morning, she was praying, I said… Lord, I put everything in Your hands and pray for the doctors, too. May He provide a [medical] team that does not even know they are God’s vessels. Everything is from God. She went to the labour ward and then the doctor came, and I told him I was the father. He asked me how I knew the baby had already been born. I have faith, I believe in God. I am at peace; when I am at peace, I know it is good. So, he said we had a boy. I am telling you, you and your wife, you are strong. You really have God on your side”. (1; m; age 35 y)*


#### 3.2.3. Preaching and the Gospel

Preaching is one of the main activities. The challenge is not only preaching the Gospel to Roma in the settlements, especially to the members of the majority, but also to priests [26]:


*"I met a longtime priest. He drinks, he smokes. He has his sheep, and you know—like the shepherd, like the sheep. He greeted me and I said: ‘God’s peace, father’. I told him: ‘Father, do you know God’s truth? Do you know what a letter to the Romans (1 Cor, 3:17) says? If any man destroys the temple of God, God shall destroy him. You are a temple, and Jesus wants to dwell in your heart. If there was smoke in your parish and if there was fire, would you go there? You would not stay there; you would automatically come out. Because there is smoke; how can you destroy that temple? After all, everything that comes from the heart is said by the mouth, isn’t it? Like you, like your sheep! There is just smoke in your heart! Do you think there is Jesus there? You may even drink, but there is the Holy Spirit in you”. (3; m; age 40 y)*


#### 3.2.4. Us and the Others

For some community members, many members of the Mission are two-faced. They appreciate the change in their behavior, especially the change in lifestyle: no alcohol and cigarettes, with a healthy, balanced, and moderate diet. However, on the other hand, they compare them to lifeless people, people who are now the right and wise ones; they say that they have stopped living [26]:


*“There is that Maranatha here, so I would say it is like a sect. Members help themselves rather than others. They have changed, especially the character. When I watch them now and I pay attention to them—they are completely different. To me, it feels like they are so disguised; it is so two-faced that who knows how they live at home and so outwardly between us, that they are the best now, but who knows what it is really like between them. They stopped drinking and smoking, which is courage, but not all of them. But they are not humans anymore, they are monsters, dead; they have stopped living. I say that when God created me, may He like me sinful too, but I live, I do not do bad things that way, I just indulge in beer when I like it, yeah… Even if only two or three gulps, but I treat myself. Or I go somewhere to meet others; for them this is already closed—the end… We are sinful for them and they are excellent people. They are the smart ones now, but they are the dead ones who stopped living”. (4; m; age 48 y)*


In addition to the multiple positives (limitation of addictions, socio-pathological phenomena, etc.) of the Mission, it is necessary to point out an area whose impacts will be visible only in the future. The Mission is responsible for working with children. During meetings and praise of adults, we witnessed children being taken aside, with one of the adult women talking to them and teaching them about the Lord, about revelations, etc.

## 4. Discussion

We compared the religiosity of the Roma community in the north-east part of the Slovak Republic at the end of the 18th century, during the reign of Empress Maria Theresa, with the present day.

We found that Roma took over the denomination of the majority. In his work, Augustini named several areas in which Roma groups resided during the period he studied them: from Transylvania to the northern parts of the Kingdom of Hungary in which he lived and worked. According to his information, the religion of the Roma is significantly linked to the denomination of the majority. In the Kingdom of Hungary in the 18th century, Catholic and Protestant faiths were the most widespread. The Roma belonged to one of them, depending on the region in which they lived [5,6].

We furthermore found that Slovak Roma had changed from a religious perspective. Earlier findings from Slovakia suggest that religious change (such as the conversion to Maranatha) leads to social change [4]. Such a religious change means a broad-based change in social habits and the behavior of individuals or of a certain group. A religious change has high potential to produce a stable, positive social change in socially excluded localities [4]. Such a change was reflected and assessed as positive by all persons involved (pastors, Roma, municipalities, non-Roma fellow citizens). The religious change has a high potential for social inclusion because in Roma converts, there is an increase in positive social skills and competences, and a decline in the social behavior modes perceived as negative by the majority society [4]. Religiosity among Slovak Roma is perceived as a key social regulator [27].

We found that religion could also have negative aspects. On the one hand, social change brings positive aspects, among other things, in consolidating community bonds, providing direction, and thus shaping a person as an individual with his or her own identity and integrity. Religion has a priceless value for human freedom, existence, and a certain degree of humility. On the other hand, we also perceive risks, even dangers, of religion which can lead to intolerance, persecution, and/or violence. Ward [28], Cremer [29], and Lorkowski [30] point to the dangers associated with devotion to faith: a paternalistic attitude, bigotry, religious dogma, or focusing on faith over deeds and the weakened critical thinking of believers.

The findings in the present study show the acceptance of the denomination of the majority by the Roma community. Traditional majority churches fail. There may be several reasons for this, ranging from structural oppression to simple indifference. Social change is possible and desired in the environment of marginalized settlements. On the other hand, it can bring significant risks whose impacts have not been mapped yet.

### 4.1. Strengths and Limitations

The main strength was that the study was conducted in the same environment in which Samuel Augustini ab Hortis lived and worked almost 250 years ago. His work was analyzed in detail and his knowledge was confronted by field data collection in the environment of Roma settlements in north-eastern Slovakia. A limitation is that our results cannot be generalized to the whole territory of Slovakia; there may be a different mission in each region, or the interest in Roma by dominant churches may vary.

### 4.2. Implications

Based on our findings, we recommend intercultural training [31] to assist professionals in the field of work of religions of poor/marginalized communities. It is important to know the rules of particular denominations. For instance, Maranatha masses usually take place on Fridays (specifically in one of the research locations), and the people refuse to go to work that day. Moreover, it is important to know that religion helps to stop unhealthy habits: smoking and drinking alcohol. Professionals should know that without pastors and evangelization, this process cannot be stopped. “Investigations of religion and health have ethical and practical implications that should be addressed by the lay public, health professionals, the research community and the clergy. Future research directions point to promising new areas of investigation that could bridge the constructs of religion and health” [7].

In the future, research may focus on examining the impact of religion on the health of the Slovak Roma, detecting changes in the social behavior of a community of believers and comparing it with an atheist community from a similar socioeconomic background.

### 4.3. Conclusions

In the past, Roma took over the faith of the majority. However, the interest of the Catholic Church in the lives of Roma in marginalized settlements in the municipalities was minimal. Therefore, smaller religious movements (like Maranatha) came to the fore and were successful in the environment of the settlements because of the declared interest in the life of their inhabitants.

## Figures and Tables

**Figure 1 ijerph-19-11645-f001:**
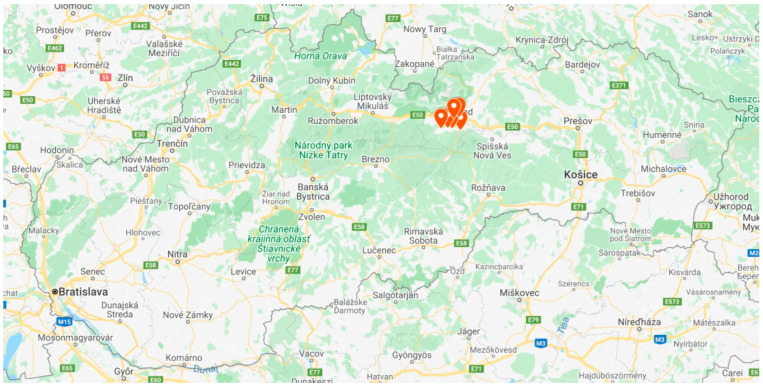
Slovakia, showing the Poprad district and research localities. Source: Google Maps, 2021 and the authors.

**Figure 2 ijerph-19-11645-f002:**
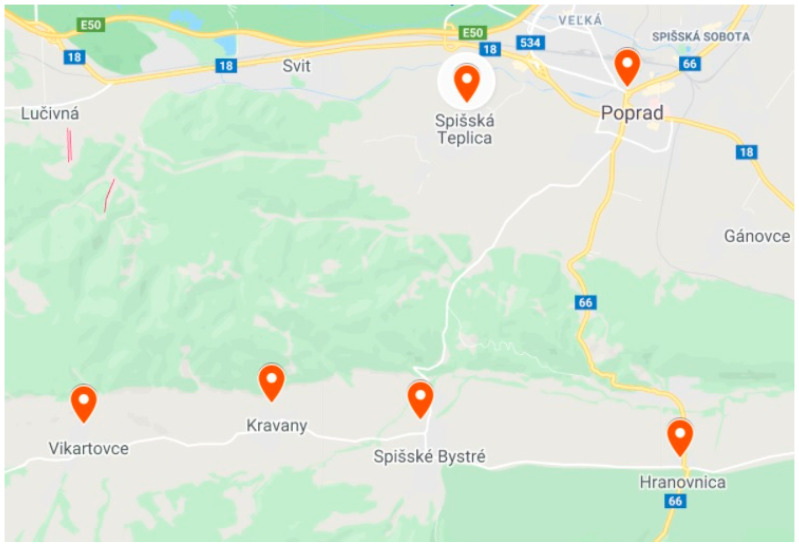
Data collection sites: integrated: Poprad, Spisska Teplica; separated: Vikartovce, Kravany; segregated: Hranovnica, Spisske Bystre. Source: Google Maps, 2021 and the authors.

**Figure 3 ijerph-19-11645-f003:**
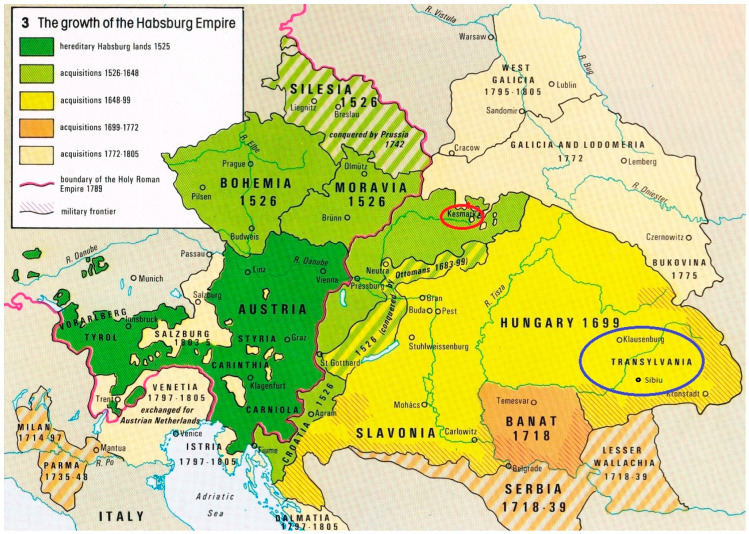
The Growth of the Habsburg Empire (the red circle denotes where Augustini worked and lived; the blue circle indicates a part of Transylvania which he describes). Source: Kolbe, 2021 [25].

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
