# Peer review of "Roma Religion: 1775 and 2018 Compared over Time"

_ijerph, 2022, doi:10.3390/ijerph191811645_

Round 1
Reviewer 1 Report
The article compares the religiosity of the Roma during the reign of
the empress Maria Theresa with the current period. The text provides
summary information about the life of the Roma national minority in
Austria-Hungary in the 18th century. I appreciate the effort to conduct
the research on this topic. I recommend to add:
- connecting the topic with the field of public health
- information on classification and conceptualization of data
- revise the Implications section in more detail
Author Response
Comments R1 on
Manuscript ID: ijerph-1861559;
Title: ’Roma Religion: 1775 and 2018 Compared over Time’
with point-by-point responses from the authors
R1.1
The article compares the religiosity of the Roma during the reign of the empress Maria Theresa with the current period. The text provides summary information about the life of the Roma national minority in Austria-Hungary in the 18th century. I appreciate the effort to conduct the research on this topic.
Response
Thank you for your very nice comment.
According to Submission Guidelines, in the topic "Conclusions": The authors "should state clearly the main conclusions and provide an explanation of the importance and relevance of the study reported". As I commented earlier, the authors make comparisons with other studies in the conclusions. This should be done in the Discussion. I suggest reviewing the conclusions.R1.2
I recommend to add:
- connecting the topic with the field of public health
Response
We added to the Introduction section:
“The relationships between religion and the health of individuals and populations has become increasingly visible in the social, behavioral, and health sciences. Recent research has validated the multidimensional aspects of religious involvement and investigated how religious factors operate through various bio-behavioral and psychosocial constructs to affect health status through proposed mechanisms that link religion and health.” [7]
Reference
[7] Chatters, M. L. Religion and Health: Public Health Research and Practice. Annual Review of Public Health 2000, 21, 335-367.
R1.3
- information on classification and conceptualization of data.
Response
We rewrote all the records obtained from the semi-structured interviews. Afterwards, the transcriptions were analyzed through an open coding process. From the codes, four main categories were created: 1. “Change of life – acceptance of Jesus”, 2. “Miracles and testimonies”, 3. “Preaching and the Gospel”, 4. “Us and the others”. We tried to present them through personal statements of the participants. In our point of view, this helps to present a true image of the daily routine of the settlement dwellers.
R1.4
- revise the Implications section in more detail
Response
We added to the Implication section:
Based on our findings, we recommend intercultural training [31] assisting professionals in the field of work of religions of poor/marginalized communities.
Moreover, it is important to know that religion helps to stop unhealthy habits: smoking and drinking alcohol. Professionals should know that without pastors and evangelization this process cannot be stopped. “Investigations of religion and health have ethical and practical implications that should be addressed by the lay public, health professionals, the research community and the clergy. Future research directions point to promising new areas of investigation that could bridge the constructs of religion and health.” [7]
- Plaza del Pino FJ, Arrogante O, Gallego-Gómez JI, Simonelli-Muñoz AJ, Castro-Luna G, Jiménez-Rodríguez D. Romani Women and Health: The Need for a Cultural-Safety Based Approach. 2022; 10(2):271.

Reviewer 2 Report
Thanks for this research, telling a lot about how minorities integrated their beliefs in European cultures. That was very interesting, generally speaking.
The way you categorized the data to identify the phenomena and develop the characteristics and dimensions of categories is also very smart, even if it is based on discursive data and on previous Augustini’s research.
I wish that the influence of nowadays foreign Missionaries would have been analyzed with higher scrutiny.
Thanks.
Author Response
Comments R2 on
Manuscript ID: ijerph-1861559;
Title: ’Roma Religion: 1775 and 2018 Compared over Time’
with point-by-point responses from the authors
R.2.1
Thanks for this research, telling a lot about how minorities integrated their beliefs in European cultures. That was very interesting, generally speaking.
Response
Thank you very much for these kind words. We really appreciate it.
R.2.2
The way you categorized the data to identify the phenomena and develop the characteristics and dimensions of categories is also very smart, even if it is based on discursive data and on previous Augustini’s research.
Response
Thank you for this comment. We tried to analyze Augustini’s work by the content analysis. Many of his descriptions completely changed in the present. Thus, we tried to approximate the daily routine of the Roma through their own words and to build four main categories through the open coding process.
R.2.3
I wish that the influence of nowadays foreign Missionaries would have been analyzed with higher scrutiny. Thanks.
Response
We realized the limitations of our study. On the other hand, we chose an independent way of a comparison between two periods of time, because we are confident that this choice helps to approximate the daily routine of the Roma in a better way.

Reviewer 3 Report
Please include for this publication the year of publication: 2021
Zachar Podolinska, T. Traditional Romani Christianity vs Pentecostal and neo-Protestant Christianity: A grounded picture of religiosity and spirituality among the Roman in the twenty-first century in Slovakia. Rom ani Studies 2021, 31, 5, 2, 155-189.
Author Response
Comments R3 on
Manuscript ID: ijerph-1861559;
Title: ’Roma Religion: 1775 and 2018 Compared over Time’
with point-by-point responses from the authors
R.3.1
Please include for this publication the year of publication: 2021
Zachar Podolinska, T. Traditional Romani Christianity vs Pentecostal and neo-Protestant Christianity: A grounded picture of religiosity and spirituality among the Roman in the twenty-first century in Slovakia. Romani Studies 2021, 31, 5, 2, 155-189.
Response
Thank you. We have added the year (2021) to the reference list.

Round 2
Reviewer 1 Report
The authors have edited the post according to review comments and we no longer have any reservations about it.